# Feedforward Compensation of Railway Static Power Conditioners in a V/v Traction Power Supply System

**Yaoguo Li** [1], **Jiaxi Hu** [2], **Zhaohui Tang** [1,*], **Yongfang Xie** [1] and **Fangyuan Zhou** [2]

[1] School of Automation, Central South University, Changsha 410083, China; Liyaog@csu.edu.cn (Y.L.);
yfxie@csu.edu.cn (Y.X.)
[2] Zhuzhou CRRC Times Electric Co., Ltd., Zhuzhou 412001, China; hujx@csrzic.com (J.H.);
zhoufy@csrzic.com (F.Z.)
* Correspondence: zhtang@csu.edu.cn

**Abstract:** Railway static power conditioners (RPC) usually improve the power quality of traction power supply systems only according to the active power of the load, which leads to inaccurate compensation. There are two factors that restrict the performance of RPC, one of which is the reactive power of the load, and the other is the system error. In order to eliminate the compensation error, a compensation optimization method is proposed. First, calculate the reactive power compensation value for the reactive power of the load. Second, introduce the amplitudes and phases of the primary currents of the V/v transformer as references for the compensation error caused by the system loss and then use fuzzy control to optimize compensation. The compensation method proposed in this paper is actually a feedforward control. In addition, this method balances the three-phase currents and enables RPC to be used in railway power supply systems with low locomotive power factors. The effectiveness of the method proposed in this paper has been confirmed by the simulation results.

**Keywords:** railway static power conditioners (RPC); V/v transformer; fuzzy control; feedforward control; balance compensation

## 1. Introduction

Electrified railways have been relatively mature in structure and application. However, power quality problems existing in the traction power supply system have been hindering the development of electrified railways. Because the three-phase V/v transformer has the advantages of a simple structure, convenient installation and maintenance and easy control, it is widely used in the traction power supply systems of electrified railways. Traction power supply systems using three-phase V/v transformers, have problems including negative sequences, harmonics, and low power factor [1–3]. To solve these problems, researchers have proposed a variety of solutions. The solutions currently applied to traction power supply systems include the following:

- Passive filters are simple and low-cost, which can filter out specific harmonics and compensate for certain reactive power, but there is a risk of overcompensation and resonance [4,5];
- Static var compensator (SVC) can realize dynamic reactive power compensation, but it is easy to generate harmonic currents [6];
- The static synchronous compensator (STATCOM) can make up for the shortcomings of SVC. Among them, single-phase STATCOMs can be applied to traction feeders with low isolation requirements, but they cannot solve the negative sequence problem. The three-phase STATCOM can dynamically compensate for the negative

> sequence of reactive power, but it needs to be connected to the 110 kV/220 kV side effectively with the help of a large-capacity, three-phase, step-down transformer [7,8];

- An active power quality compensator (APQC) requires the use of balance transformers in transformers, which makes it not widely used [9];
- Railway static power conditioners (RPC) can solve power quality problems including negative sequence, harmonics and low power factor [10–12];
- A hybrid railway power compensator (HRPC) has multiple structures. The compensation principle of the regulator is to use two compensators at the same time, such as combining RPCs with a passive filter [13] or RPC with SVC [14,15], etc. Compared with RPCs, HRPCs do have advantages. However, in the existing research or engineering applications, most of them use a passive branch to bear the fixed compensation. There is no clear coordinated scheme for an RPC and passive branch compensation allocation, and its control is more complex, which is not conducive to the expansion and transformation of the system. Moreover, the traction load has strong time variance and randomness, and there are still overcompensation and resonance risks under some working conditions. In fact, this method weakens the flexibility of RPCs.

Traction power supply systems based on three-phase V/v transformers widely use RPCs to improve power quality. The RPC was first proposed by [16,17]. At the beginning, the structure of RPCs contained two H-bridge converters connected back-to-back by a common DC capacitor (supporting capacitor). With the development of electrified railways, multiple extension structures with RPCs have been explored. In reference [12], an RPC circuit of the full bridge structure is shown, and its control is relatively simple. In reference [18], an RPC circuit is constructed based on a half-bridge circuit, which saves hardware and reduces losses compared with a full bridge but requires two supporting capacitors in series, so the voltage balance of the capacitors needs to be maintained, and furthermore, the control is more complicated. With the increase of the rated capacity of RPCs, the structures of RPCs are various. For instance, it is possible to connect full-bridge and or half-bridge circuits in series [19] or parallel [9] to form multiple structures, but their compensation principles are basically the same. Since this article aims to study the optimization problem of compensation, for the convenience of research, this article uses a full-bridge RPC circuit and a V/v transformer in the traction power supply system.

In the calculation of the RPC compensation reference value, almost all researchers have adopted the same viewpoint. In reference [12,20], the causes of power quality problems such as the negative sequence harmonics of the V/v traction power supply system are analyzed, the calculation method of the compensation current reference signal is given and its control algorithm uses hysteresis for compensation control. Reference [21] uses the same compensation current reference signal calculation method as [12], but the control algorithm uses repeated prediction and deadbeat control. Many scholars have conducted in-depth research on the control algorithm of RPC, including fuzzy control, recursive PI (proportion integral) control and parallel quasi-resonant control. As the load borne by the traction network continues to increase, the kilovolt-ampere (kVA) rating of RPC also increases, and the cost increases accordingly. In order to reduce the kVA rating of RPC and cost, the reference [15] adopts the method of combining RPC and TSC (thyristor switched capacitor) such that TSC bears part of the compensation current. In order to improve the compensation performance, reference [22] combines MSVC (magnetic SVC) with RPC, but their calculations of the compensation current reference signal are basically the same. They all assume that the power factor of the load is close to 1, so the reactive power of the load is not considered. The above control strategy can be classified into the maximum compensating mode, which requires the primary side of the V/v transformer to meet the following conditions:

$$\begin{cases} NSC = 0 \\ NSV = 0 \\ PF = 1 \\ THD = 0 \end{cases}, \tag{1}$$

where *NSC*, *NSV*, *PF*, and *THD* represent negative sequence current, negative sequence voltage, power factor and total harmonic distortion, respectively.

Reference [23] proposed the FrCM (flexible fractional compensating mode) method; this method does not need to add additional circuit components, and the goal of compensation satisfies Equation (1) as much as possible. In fact, the reference [23] relaxes the compensation conditions to reduce kVA. Because there is no complete compensation, negative sequence current and harmonics still exist, and this compensation mode is called optimal compensation. In the calculation of the compensation current, although the reactive current of the load is considered, the reference current in the actual calculation is consistent with the full compensation mode. The difference is that according to the optimization goal, the compensated current is smaller than the full compensation mode. The current does not compensate for the reactive power of the load and system loss. In fact, the power factor of the load fluctuates, and in some cases, is even very low. If the system loss is not considered, the optimization goal may not be as expected, and the optimization compensation is not meaningful. Therefore, a more comprehensive calculation is needed to obtain an accurate compensation amount.

This article proposes optimization and improvement on the general compensation scheme. First, for loads that may have reactive power, feedforward compensation is based on the reactive power of the load current. Second, for the possible compensation errors and transformer losses in the system, the amplitudes and phases of V/v transformer primary currents are introduced. In addition, the fuzzy control is used to optimize the system compensation, which optimizes the compensation amount more accurately and reduces the tracking error. This method improves the power quality indexes of the traction power supply system on the basis of ensuring the stability of the system. Moreover, this method can be applied not only to high-speed railway traction systems but also to ordinary electrified railways, which improves the universality of RPC and makes the traction power supply system safer and more reliable. This method only needs to improve the algorithm on the basis of the original RPC without adding any additional circuit components.

This article contains the following parts: Section 2 briefly describes the system structure and analyzes the researched problems. Section 3 proposes detailed solutions. Section 4 uses the proposed method for simulation verification through the analysis and comparison of the simulation results of the general compensation method, Section 5 draws the conclusion of this article.

## 2. Materials and Methods

### 2.1. System Structure

The structure of an RPC with three-phase V/v transformers is depicted in Figure 1. The V/v transformer is connected to the 110 kV three-phase power grid, which is converted into two 27.5 kV single-phase AC voltage sources through the V/v transformer. For convenience, the right side is named $\alpha$ phase power supply arm, which is connected to Phase A and Phase C through the transformer, and the left side is named $\beta$ phase power supply arm, which is connected to Phase B and Phase C through the transformer. The two power supply arms are connected to different traction networks to supply power to the locomotive, which is the load. The RPC, consisting of two converters that are connected back-to-back through a DC capacitor with a stable DC-link voltage, is connected to the two power supply arms via two step-down transformers. In fact, the converter is a full-

bridge circuit, so the two converters are usually connected to the single-phase step-down transformers via two output inductors $L_\alpha$ and $L_\beta$, respectively. The role of the two inductors is to buffer sudden changes and filter in current. The RPC can be controlled as a current source to convert a certain amount of active power from one power supply arm to another. Because the structure of the RPC is symmetrical, this kind of conversion is bidirectional. The RPC can also compensate for reactive power and suppress harmonics. Therefore, the RPC can solve the power quality problems of traction power supply systems.

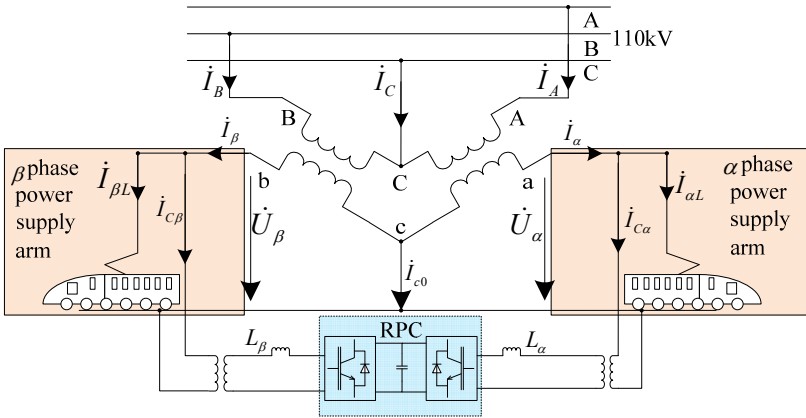

**Figure 1.** Traction power system with a three-phase V/v transformer and a railway static power conditioners (RPC).

### 2.2. Operating Principles of an RPC

For the convenience of calculation and analysis, assume that the transformation ratio of all transformers in the system is 1.

As shown in Figure 1, assume that the three-phase primary voltages of V/v transformer are

$$
\begin{cases}
U_A(t) = sin(\omega t) \\
U_B(t) = sin(\omega t - \dfrac{2\pi}{3}) \, , \\
U_C(t) = sin(\omega t + \dfrac{2\pi}{3})
\end{cases}
\tag{2}
$$

where $\omega$ is frequency of the power system. Therefore, the load current $I_{\alpha L}$ of $\alpha$ phase power supply arm and the load current $I_{\beta L}$ of $\beta$ phase power supply arm can be expressed as follows:

$$
\begin{cases}
I_{\alpha L}(t) = I_{\alpha Lf}sin(\omega t - \dfrac{\pi}{6} + \varphi_\alpha) \\
I_{\beta L}(t) = I_{\beta Lf}sin(\omega t - \dfrac{\pi}{2} + \varphi_\beta)
\end{cases} \, ,
\tag{3}
$$

where $\varphi_\alpha$ and $\varphi_\beta$ are the phase angles of the current and voltage of the two power supply arms, respectively. $I_{\alpha Lf}$ and $I_{\beta Lf}$ are the amplitudes of the load currents $I_{\alpha L}$ and $I_{\beta L}$, respectively. In previous studies, researchers have regarded loads as pure resistive loads, so they have assumed that $\varphi_\alpha = \varphi_\beta = 0$ [10,12,15,18,21]. This assumption is temporarily adopted in the following analysis.

The phasor diagram of the system without compensation is shown in Figure 2a. $I_A$, $I_B$ and $I_C$ are the primary currents of the V/v transformer, and their directions are shown in

Figure 1. Likewise, $I_\alpha$, $I_\beta$ and $I_{C0}$ are the secondary currents of the V/v transformer, and $U_\alpha$ and $U_\beta$ are the voltages of the two power supply arms.

As depicted in Figure 2a, $I_A$, $I_B$ and $I_C$ are asymmetric, which injects large quantities of negative sequence currents into the power grid, and they are out of phase with the phase voltage ($U_A$, $U_B$, $U_C$), which causes a low power factor [2,3,6].

In order to make $I_A$, $I_B$ and $I_C$ balanced, it is necessary to control RPC to shift active power and compensate for reactive power.

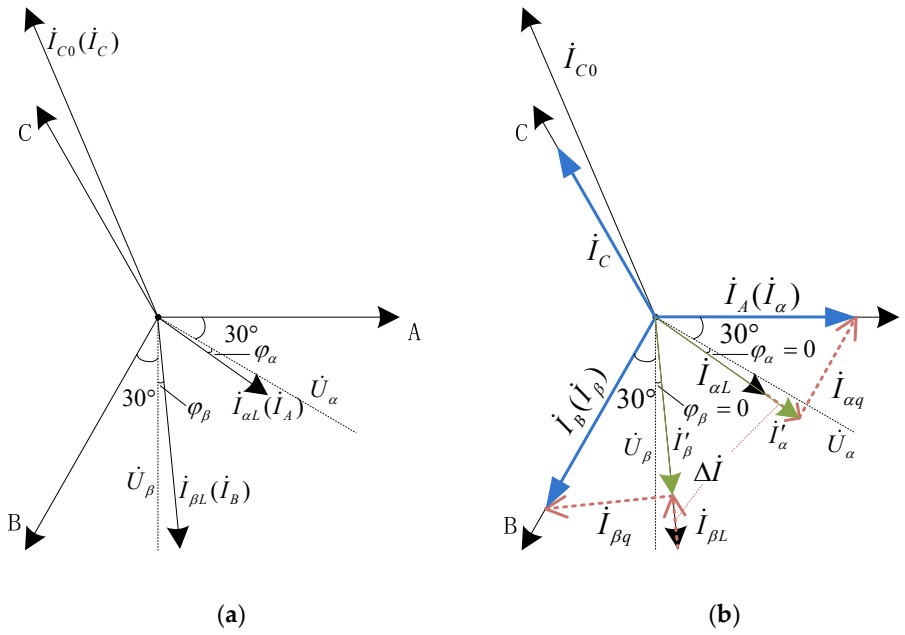

(**a**)　　　　　　　　　　　　　　　　(**b**)

**Figure 2.** Phasor diagram of the V/v traction power supply system. (**a**) Without RPC compensation; (**b**) After RPC compensation. (In order to show the existence of $\varphi_\alpha$ and $\varphi_\beta$, although it is assumed that $\varphi_\alpha = \varphi_\beta = 0$, $U_\alpha$ ($U_\beta$) and $I_{\alpha L}$ ($I_{\beta L}$) are not overlapped in the figure. In fact, under this assumption, $U_\alpha$ ($U_\beta$) and $I_{\alpha L}$ ($I_{\beta L}$) are overlapped.)

As shown in Figure 2b, assuming that the load power of $\alpha$ is heavier than $\beta$, the RPC converts half of the current difference of the two load currents from $\alpha$ phase power supply arm to $\beta$ phase power supply arm. $\Delta I$ is the current of active power converted by RPC, which is calculated as follows:

$$\Delta I = \frac{1}{2}(I_{\alpha Lf} + I_{\beta Lf}) . \tag{4}$$

The currents of the two power supply arms are denoted as $I'_\alpha$ and $I'_\beta$ after converting, respectively. The amplitudes of $I_\alpha$ and $I_\beta$ are equal. They can be expressed as:

$$\begin{cases} I'_\alpha(t) = I_{mp} sin(\omega t - \dfrac{\pi}{6}) \\ I'_\beta(t) = I_{mp} sin(\omega t - \dfrac{\pi}{2}) \end{cases}, \tag{5}$$

where

$$I_{mp} = I_{\alpha Lf} - \Delta I = I_{\beta Lf} + \Delta I . \tag{6}$$

next, the RPC adds reactive currents $I_{\alpha q}$ and $I_{\beta q}$ to $\alpha$ phase power supply arm and $\beta$ phase power supply arm. Since $I_{\alpha q}$ leads $I'_\alpha$ by $\pi/2$ and $I_{\beta q}$ lags $I'_\beta$ by $\pi/2$, $I_{\alpha q}$ and $I_{\beta q}$ meet the following conditions:

$$\begin{cases} I_{\alpha q}(t) = I_{mp} tan(\frac{\pi}{6})cos(\omega t - \frac{\pi}{6}) \\ I_{\beta q}(t) = I_{mp} tan(\frac{\pi}{6})cos(\omega t - \frac{\pi}{2}) \end{cases}. \tag{7}$$

The total currents of two power supply arms are $I_\alpha$ and $I_\beta$ after compensation, and the phase angles of $I_\alpha$ and $I_\beta$ are in phase with Phase A voltage and Phase B voltage. Then

$$\begin{cases} I_\alpha = I'_\alpha(t) + I_{\alpha q}(t) = I_{mp} sin(\omega t - \frac{\pi}{6}) + I_{mp} tan(\frac{\pi}{6})cos(\omega t - \frac{\pi}{6}) \\ I_\beta = I'_\beta(t) + I_{\beta q}(t) = I_{mp} sin(\omega t - \frac{\pi}{2}) + I_{mp} tan(\frac{\pi}{6})cos(\omega t - \frac{\pi}{2}) \end{cases}, \tag{8}$$

hence, the total compensating currents of RPC can be calculated:

$$\begin{cases} I_{c\alpha} = I_\alpha(t) - I_{\alpha L}(t) \\ I_{c\beta} = I_\beta(t) - I_{\beta L}(t) \end{cases}. \tag{9}$$

In fact, $I_{c\alpha}$ and $I_{c\beta}$ are the current references for RPC.

According to instantaneous reactive power theory [24], Equation (9) can be implemented as shown in Figure 3.

PLL in Figure 3 refers to a phase-locked loop. The PLL can get the frequency and phase of the input signal. $u_\alpha$ and $u_\beta$ are the voltages of the two power supply arms, respectively.

LPF in Figure 3 refers to the low-pass filter, and Shifter is the phase shifter.

Reference [12] theoretically derives the correctness of Figure 3, and it also draws a conclusion that the method can not only implement Equation (9) but also give a reference current to suppress harmonics.

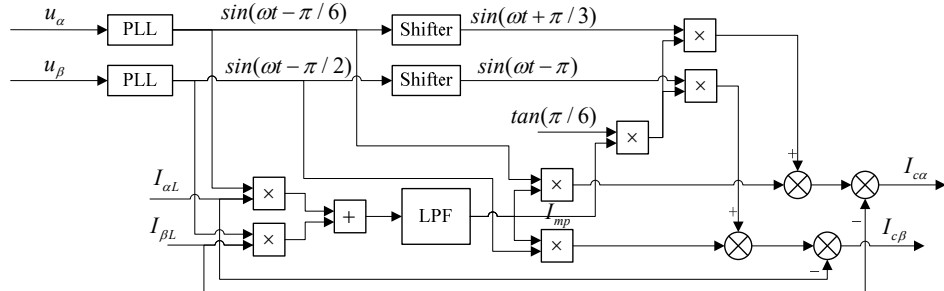

**Figure 3.** The implementation of current references for RPC.

### 2.3. Control Strategy of RPC

RPC actually realizes energy conversion through the rectification and inverter process, and the supporting capacitor is the bridge of this process. In this process, the voltage of the supporting capacitor should be kept stable [10,12,20]. Therefore, most researchers adopt dual-loop control. The outer loop controls the voltage of the DC capacitor to be constant, and the inner loop controls the compensation current. The control strategy is illustrated in Figure 4.

$U_{ref}$ in Figure 4 is the voltage reference value of the DC capacitor, and $U_{DC}$ is its actual value. $I_{C\alpha}$ and $I_{C\beta}$ in Figure 4 are the actual values of the compensation currents of the two power supply arms. The main function of the mean filter in Figure 4 is to filter out the ripples of $U_{DC}$ to avoid generating harmonics [10].

Although the basic structure of most control strategies is shown in Figure 4, the difference is that the control algorithms are diverse. Each algorithm has advantages and

disadvantages. In fact, it is a trade-off between control speed and control accuracy. Considering the performance of different algorithms, the voltage outer loop adopts a composite PI control algorithm [10,20], and the current inner loop adopts a quasi-resonant control algorithm [25].

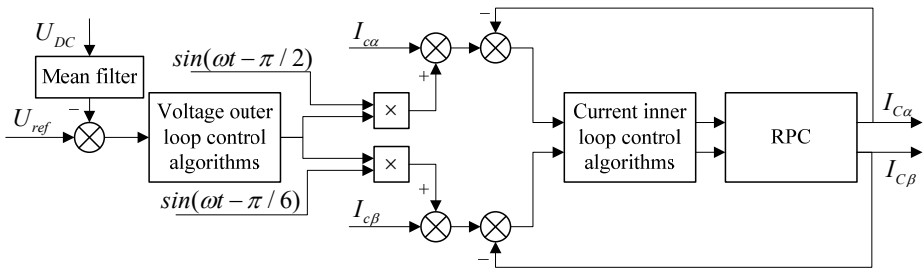

**Figure 4.** Control strategy of RPC.

### 2.4. Problem Statement

It is easy to observe that the primary side currents of a V/v transformer ($I_A$, $I_B$, and $I_C$) are symmetric three-phase currents, and $I_A$, $I_B$ and $I_C$ are in phase with the phase voltage ($U_A$, $U_B$ and $U_C$) based on the above analysis. The reactive power of the loads and system loss are ignored in previous research but exist in the actual traction power supply system.

The reactive power of the loads and transformer loss will cause the current references to be inaccurate. Many researchers use Equation (9) as the reference currents of the control system to continuously optimize the performance of the control algorithms so that $I_{C\alpha}$ ($I_{C\beta}$) can quickly follow $I_{c\alpha}$ ($I_{c\beta}$) with very small errors. However, the inaccuracy of the reference current will result in the power supply system always being in error with the expectation, which is not a better result.

Assume that the power factors of the two power supply arms are $cos\varphi_\alpha$ and $cos\varphi_\beta$, respectively. Equation (3) can be expressed as follows:

$$\begin{cases} I_{\alpha L}(t) = I_{\alpha Lf}sin(\omega t - \dfrac{\pi}{6} + \varphi_\alpha) + \sum_{h=2}^{\infty} I_{\alpha h}sin(h\omega t + \varphi_{\alpha h}) \\ I_{\beta L}(t) = I_{\beta Lf}sin(\omega t - \dfrac{\pi}{2} + \varphi_\beta) + \sum_{h=2}^{\infty} I_{\beta h}sin(h\omega t + \varphi_{\beta h}) \end{cases} \tag{10}$$

where $I_{\alpha h}sin(h\omega t + \varphi_{\alpha h})$ and $I_{\beta h}sin(h\omega t + \varphi_{\beta h})$ are *h*th-order harmonic currents of the two-power supply arm. Multiplication of $I_{\alpha L}$ and $I_{\beta L}$ with the synchronous voltages of the two power supply arms is used to obtain the instantaneous powers $P_\alpha$ and $P_\beta$:

$$\begin{cases} P_\alpha = I_{\alpha L}(t)sin(\omega t - \dfrac{\pi}{6}) = \dfrac{I_{\alpha Lf}cos\varphi_\alpha}{2} - \dfrac{I_{\alpha Lf}}{2}cos(2\omega t - \dfrac{\pi}{3} + \varphi_\alpha) + \sum_{h=2}^{\infty} I_{\alpha h}sin(h\omega t + \varphi_{\alpha h})sin(\omega t - \dfrac{\pi}{6}) \\ P_\beta = I_{\beta L}(t)sin(\omega t - \dfrac{\pi}{2}) = \dfrac{I_{\beta Lf}cos\varphi_\beta}{2} - \dfrac{I_{\beta Lf}}{2}cos(2\omega t - \pi + \varphi_\beta) + \sum_{h=2}^{\infty} I_{\beta h}sin(h\omega t + \varphi_{\beta h})sin(\omega t - \dfrac{\pi}{2}) \end{cases} \tag{11}$$

where $P_\alpha$ and $P_\beta$ contain a DC component and an AC component, add $P_\alpha$ and $P_\beta$ and pass through a LPF to get the sum of the DC component:

$$\overline{P}_\alpha + \overline{P}_\beta = \frac{1}{2}(I_{\alpha Lf}cos\varphi_\alpha + I_{\beta Lf}cos\varphi_\beta). \tag{12}$$

It is not difficult to infer that Equation (12) is the average value of the active component of the load current of the two power supply arms:

$$\overline{P}_\alpha + \overline{P}_\beta = \frac{1}{2}(\underbrace{I_{\alpha Lf}cos\varphi_\alpha}_{I_{\alpha Lp}} + \underbrace{I_{\beta Lf}cos\varphi_\beta}_{I_{\beta Lp}}) \,. \tag{13}$$

In effect, this is $I_{mp}$. Therefore, the compensation amount of active power ($\Delta I$) and the compensation amount of reactive power ($I_{\alpha q}$ and $I_{\beta q}$) remain unchanged. Then, the phasor diagram of the system after compensation when $\varphi_\alpha \neq 0$ and $\varphi_\beta \neq 0$ is shown in Figure 5.

As shown in Figure 5, the primary side currents of the V/v transformer ($I_A$, $I_B$, $I_C$) are still unbalanced. It can be inferred from Figure 5 that the reactive power of the load needs to be compensated for. At the same time, when considering the loss of the power supply system, this imbalance will worsen. Loss exists in many parts of the system, including transformers, RPC, etc., but it is difficult to accurately calculate the loss.

In order to eliminate the system error caused by load reactive power and system loss, two optimization methods are proposed in this paper:

1. The instantaneous reactive power theory is used to compensate for the load reactive power dynamically in real time;
2. The amplitudes and phases of $I_A$ and $I_B$ are used as references to carry out auxiliary fine-tuning compensation of the system.

This article aims to address the following issues:

1. How to eliminate subtle system deviations caused by loss.
2. Does the improvement of the system effectively improve the performance of the traction power supply system?

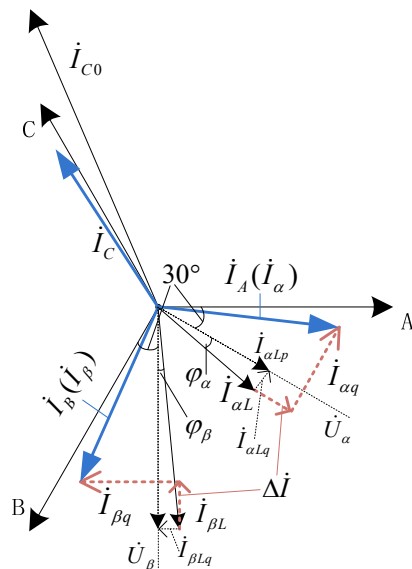

**Figure 5.** Phasor diagram of the system after compensation under actual conditions where $\varphi_\alpha \neq 0$ and $\varphi_\beta \neq 0$. (This figure shows the case of $\varphi_\alpha < 0$, $\varphi_\beta > 0$. The phasor diagram in other cases is similar to this figure. This article uses the situation in this figure to illustrate.)

## 3. Proposed Compensation Optimization Strategy

### 3.1. Load Reactive Power Dynamic Compensation Optimization

The reactive power of the load can be expressed as following:

$$\begin{cases} I_{\alpha Lq}(t) = I_{\alpha Lf}sin\varphi_\alpha cos(\omega t - \dfrac{\pi}{6}) \\ I_{\beta Lq}(t) = -I_{\beta Lf}sin\varphi_\beta cos(\omega t - \dfrac{\pi}{2}) \end{cases} \tag{14}$$

Obviously, as long as $\varphi_\alpha$ is known, the reactive power of the load can be calculated. However, since the reactive power may fluctuate according to the operating state of the locomotives, it is difficult to directly measure the phase difference between the load current and the voltage of the power supply arm. Reactive power compensation needs to be real-time. Even if the phase difference is measured, the timeliness will be very poor, and the phase difference measurement result is likely to be inaccurate. According to the instantaneous reactive power theory, multiplying the $I_{\alpha L}$ in Equation (10) with the voltage of $\alpha$ phase power supply arm leads by $\pi/2$; multiplying the $I_{\beta L}$ in Equation (10) with the voltage of $\beta$ power supply arm lags by $\pi/2$ to get:

$$\begin{cases} Q_\alpha = \dfrac{I_{\alpha Lf}sin\varphi_\alpha}{2} + \dfrac{I_{\alpha Lf}}{2}sin(2\omega t - \dfrac{\pi}{3} + \varphi_\alpha) + \sum_{h=2}^{\infty} I_{\alpha h}sin(h\omega t + \varphi_{\alpha h})sin(\omega t - \dfrac{\pi}{6} + \dfrac{\pi}{2}) \\ Q_\beta = -\dfrac{I_{\beta Lf}sin\varphi_\beta}{2} - \dfrac{I_{\beta Lf}}{2}sin(2\omega t - \pi + \varphi_\beta) + \sum_{h=2}^{\infty} I_{\beta h}sin(h\omega t + \varphi_{\beta h})sin(\omega t - \dfrac{\pi}{2} - \dfrac{\pi}{2}) \end{cases} \tag{15}$$

$Q_\alpha$ and $Q_\beta$ are used to calculate the DC component through a LPF respectively:

$$\begin{cases} \overline{Q}_\alpha = \dfrac{1}{2}I_{\alpha Lf}sin\varphi_\alpha \\ \overline{Q}_\beta = -\dfrac{1}{2}I_{\beta Lf}sin\varphi_\beta \end{cases} \tag{16}$$

Comparing Equations (16) and (13), Equation (16) obtains half of the reactive power amplitude of the load. So, load reactive power compensation can be realized as shown in Figure 6.

$I_{c\alpha Lq}$ and $I_{c\beta Lq}$ are the load reactive power compensations of the two power supply arms, respectively. Since the reactive power is calculated by the load current, the load reactive power compensation depends on the change of the load reactive current. Therefore, Figure 6 can dynamically compensate for the reactive power of the load in real time.

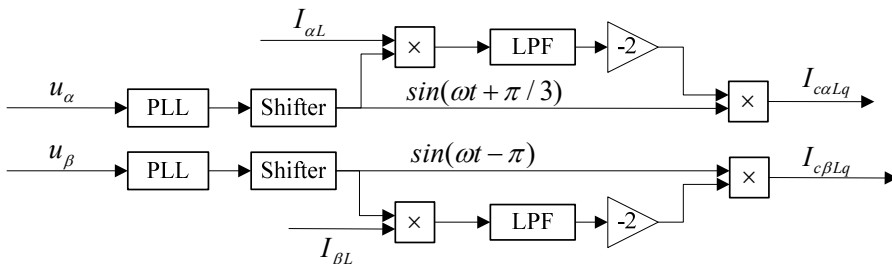

**Figure 6.** Load reactive power dynamic compensation.

### 3.2. Auxiliary Fine-Tuning Compensation of the System

In the previous research, the compensation value was determined by the load current, but the purpose of our compensation method was to balance the primary current of the V/v transformer. The primary current of the V/v transformer is affected by the loss of the system, and the calculation error of the reference current cannot be effectively

eliminated. However, these losses and errors are difficult to detect. If the compensation is accurate and error-free, the amplitudes of the primary currents $I_A$ and $I_B$ of the V/v transformer are equal, and the phases are consistent with $U_A$ and $U_B$, respectively. Therefore, according to the amplitudes and phases of $I_A$ and $I_B$ as the references of the compensation system, the closed loop of the compensation system is expanded to the primary side of the V/v transformer.

Figures 3 and 6 show block diagrams to compensate for most of the reactive power and active power of the system. In order to optimize the compensation of the system, the compensation needs to be fine-tuned. Assuming that $dI_{\alpha p}$ and $dI_{\alpha q}$ are the fine adjustments of reactive power compensation and active power compensation of the $\alpha$ phase power supply arm, $dI_{\beta p}$ and $dI_{\beta q}$ are the fine adjustments of the $\beta$ phase power supply arm reactive power compensation and active power compensation, respectively.

Assume that the phase angle and amplitude of $I_A$ are $\theta_A$ and $I_{Af}$, the phase angle and amplitude of $I_B$ are $\theta_B$ and $I_{Bf}$ and the phase angle of $U_A$ and $U_B$ are $\theta_{A0}$ and $\theta_{B0}$. Then,

$$\begin{cases} dI = I_{Af} - I_{Bf} \\ d\theta_A = \theta_{A0} - \theta_A \\ d\theta_B = \theta_{B0} - \theta_B \end{cases} , \tag{17}$$

where $d\theta_A$ and $d\theta_B$ are the phase angle difference, its sign indicates the direction of reactive power compensation, but the compensation magnitude cannot be accurately calculated. $dI$ is the current amplitude difference. If $dI$ is positive, it means $dI_{\alpha p}$ or $dI_{\beta p}$ needs to be increased. If $dI$ is negative, it means $dI_{\alpha p}$ or $dI_{\beta p}$ needs to be increased.

Traction power supply system is a nonlinear system, and it is difficult to establish an accurate mathematical model. Fuzzy control has strong robustness. The variables to be controlled have obvious fuzzy logic, and it is easy to refine fuzzy rules.

In order to achieve better compensation performance, the fuzzy control is used to fine-tune the compensation.

The fuzzy sets $E_A$, $E_B$, $E_I$, $U_{AP}$, $U_{AQ}$, $U_{BP}$ and $U_{BQ}$ of $d\theta_A$, $d\theta_B$, $dI$, $dI_{\alpha p}$, $dI_{\alpha q}$, $dI_{\beta p}$ and $dI_{\beta q}$ are {PB PS 0 NS NB}, and their membership function uses sigmoid function.

According to the above analysis, the fuzzy rules of $U_{AP}$, $U_{AQ}$, $U_{BP}$ and $U_{BQ}$ are presented in Tables 1–3.

**Table 1.** Fuzzy rules for $dI_{\alpha p}$ and $dI_{\beta p}$.

| Set | Level | | | | |
|---|---|---|---|---|---|
| $E_I$ | PB | PS | 0 | NS | NB |
| $U_{AP}$ | NB | NS | 0 | 0 | 0 |
| $U_{BP}$ | 0 | 0 | 0 | NS | NB |

**Table 2.** Fuzzy rules for $dI_{\alpha q}$.

| $E_A$ | $E_I$ | | | | |
|---|---|---|---|---|---|
| | **PB** | **PS** | **0** | **NS** | **NB** |
| PB | PS | PS | PB | PB | PB |
| PS | 0 | 0 | PS | PS | PS |
| 0 | NS | 0 | 0 | 0 | 0 |
| NS | NB | NS | NS | NS | NS |
| NB | NB | NB | NB | NB | NB |

**Table 3.** Fuzzy rules for *dIβq*.

| $E_B$ | $E_I$ | | | | |
|---|---|---|---|---|---|
| | **PB** | **PS** | **0** | **NS** | **NB** |
| NB | PB | PB | PB | PS | PS |
| NS | PS | PS | PS | 0 | 0 |
| 0 | 0 | 0 | 0 | 0 | NS |
| PS | NS | NS | NS | NS | NB |
| PB | NB | NB | NB | NB | NB |

The defuzzification algorithm uses the gravity method [26,27]. The gravity method is widely used in defuzzification. It is a reasonable and explanatory algorithm. Then, we can obtain:

$$
\begin{cases}
dI_{\alpha p}(k) = dI_{\alpha p}(k-1) + \dfrac{\sum_{i=1}^{m} \mu_{U_{AP}}(E_{Ii}) U_{BPi}}{\sum_{i=1}^{m} \mu_{U_{AP}}(E_{Ii})} \\[4mm]
dI_{\beta p}(k) = dI_{\beta p}(k-1) + \dfrac{\sum_{i=1}^{m} \mu_{U_{BP}}(E_{Ii}) U_{BPi}}{\sum_{i=1}^{m} \mu_{U_{BP}}(E_{Ii})} \\[4mm]
dI_{\alpha q}(k) = dI_{\alpha q}(k-1) + \dfrac{\sum_{i=1}^{m} \mu_{U_{AQ}}(E_{Ii}, U_{AQi}) U_{AQi}}{\sum_{i=1}^{m} \mu_{U_{AQ}}(E_{Ii}, U_{AQi})} \\[4mm]
dI_{\beta q}(k) = dI_{\beta q}(k-1) + \dfrac{\sum_{i=1}^{m} \mu_{U_{BQ}}(E_{Ii}, U_{BQi}) U_{BQi}}{\sum_{i=1}^{m} \mu_{U_{BQ}}(E_{Ii}, U_{BQi})}
\end{cases}, \tag{18}
$$

where (*k*−1) presents the previous moment, and *dIαp*(0) = *dIαq*(0) = *dIβp*(0) = *dIβq*(0) = 0.

Figures 5 and 6 represent the main compensation system. Auxiliary fine tuning before the main compensation system is stable will cause the reference current calculation error and affect the system stability. Therefore, the fine tuning needs to be performed after the main compensation system is stable. The stable condition of the main compensation system is judged by the output value of PF, and the conditions for putting in auxiliary fine-tuning are:

$$
\left| PF(T_i) - PF(T_i - 1) \right| < \delta, i = 1, 2, 3, 4, 5, \tag{19}
$$

where *Ti* is the sampling period and *δ* is the threshold of a stable condition. Equation (19) indicates that the change of PF for 5 consecutive sampling periods is less than the threshold.

The conditions for casting auxiliary fine-tuning are:

$$
\left| PF(T_i) - PF(T_i - 1) \right| > \xi, \tag{20}
$$

where *ξ* is the unstable condition threshold.

The overall control strategy of the system is shown in Figure 7.

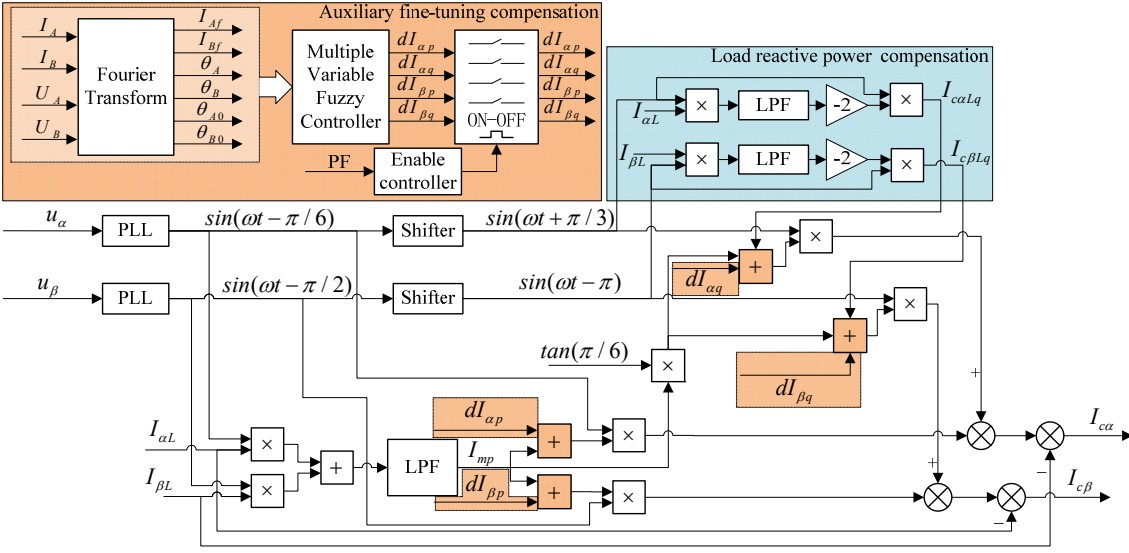

**Figure 7.** Overall control strategy of the system.

## 4. Simulation Analysis and Verification

In order to verify the effectiveness of the proposed method, simulations were carried out in MATLAB. The main simulation parameters of the system are shown in Table 4. The DC capacitor of 20,000 μF at a voltage of 2000 V was adopted in the simulation system. However, in practical engineering applications, RPC usually adopts multiple structures, so the capacity and voltage level of the DC capacitor in the actual system were much smaller [9,19].

**Table 4.** Simulation parameters of the system.

| Parameter | Value |
|---|---|
| Grid voltage | 110 kV |
| V/v transformer ratio | 110:27.5 |
| Step-down transformer ratio for RPC | 27.5:1 |
| Output reactance | $L_\alpha = L_\beta = 1.5$ mH |
| DC capacitor | 20,000 μF |
| Voltage of DC capacitor | $U_{ref} = 2000$ V |

In order to simulate the active power and reactive power of the locomotive, a combination of resistance, inductance and capacitance were used as a load to connect to the 27.5 kV power supply arm, and an uncontrollable rectifier load was used to simulate harmonic current [12].

At the same time, in order to simulate the system loss, a pure resistance load was connected in the $\beta$ supply arm, and the current consumed was not counted in the $I_{\beta L}$, as shown in Figure 8. The power of system loss is $P_l$.

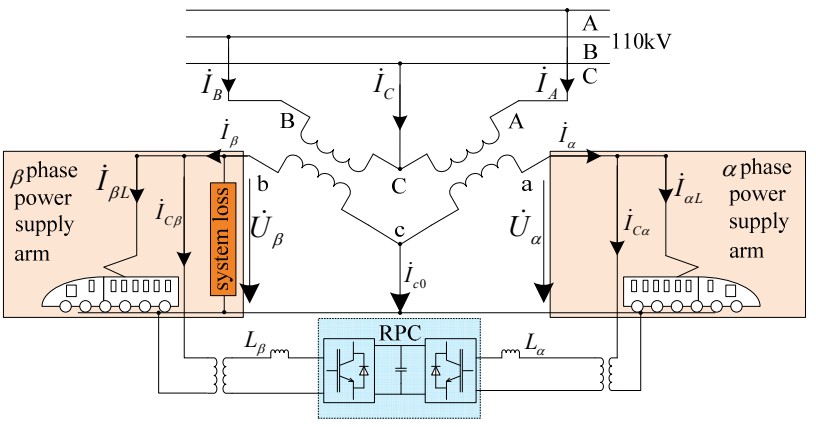

**Figure 8.** System loss $P_l$.

In order to verify the performance of the fuzzy controller, we designed a simple PI controller as a comparison. The block diagram of the PI controller is shown in Figure 9. At the same time, a method without compensation was used as a comparison [12].

The simulation was carried out in three different load situations, and the load parameters are shown in Table 5. From 0 s to 0.1 s, the system was initialized, and the DC capacitor was charged to $U_{DC} = 2000$ V.

The simulation results are shown in Figures 10–12.

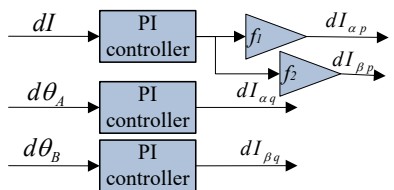

**Figure 9.** A simple PI controller that compensates for RPC.

$f_1$ and $f_2$ in Figure 9 were the compensation factors. They were determined by experience, and their signs were the opposite.

**Table 5.** Simulation parameters of load.

| Case | Intervals (s) | Active Power of $\alpha$-Phase (kW) | Reactive Power of $\alpha$-Phase (kVar) | Active Power of $\beta$-Phase (kW) | Reactive Power of $\beta$-Phase (kVar) | $P_l$ (kW) | Compensation Scheme |
|---|---|---|---|---|---|---|---|
| Case 1 | 0.1–0.2 | | | | | | System without RPC |
| | 0.2–0.3 | | | | | | RPC without compensation |
| | 0.3–0.4 | 1000 | 500 | 4800 | −500 | 0 | RPC with PI controller |
| | 0.4–0.5 | | | | | | Compensation scheme proposed |
| Case 2 | 0.1–0.2 | | | | | | System without RPC |
| | 0.2–0.3 | | | | | | RPC without compensation |
| | 0.3–0.4 | 0 | 0 | 0 | 0 | 0 | RPC with PI controller |
| | 0.4–0.5 | | | | | | Compensation scheme proposed |

| | | | | | | |
|---|---|---|---|---|---|---|
| **Case 3** | 0.1–0.2 | | | | | | System without RPC |
| | 0.2–0.3 | 1000 | 500 | 4800 | −500 | 500 | RPC without compensation |
| | 0.3–0.4 | | | | | | RPC with PI controller |
| | 0.4–0.5 | | | | | | Compensation scheme proposed |

If the reactive power is positive, it means inductive reactive power; otherwise, capacitive reactive power.

As shown in Figure 10, PF was low due to the reactive power of the load. $dI$, $d\theta_A$, $d\theta_B$ and total reactive power Q of the supply system are still relatively large, although uncompensated-for RPC can increase PF. Using a PI controller to compensate for RPC can effectively reduce $dI$, $d\theta_A$, $d\theta_B$ and Q, but the proposed method has a better compensation effect. The method proposed in this paper reduces $dI$ to less than 1 A, and the phase difference between voltage and current is also reduced to less than 1 degree.

As shown in Figure 10, there was no load current due to no load in the system. The uncompensated RPC did not work, but the system had losses, $I_A$, $I_B$ and $I_C$ were still unbalanced and PF was very low. Using a PI controller to compensate for RPC can improve its compensation effect, but the effect improvement is not perfect. The phase difference between voltage and current is still large. The method presented in this paper can effectively solve these problems and reduce the phase differences between voltage and current to less than 2 degrees.

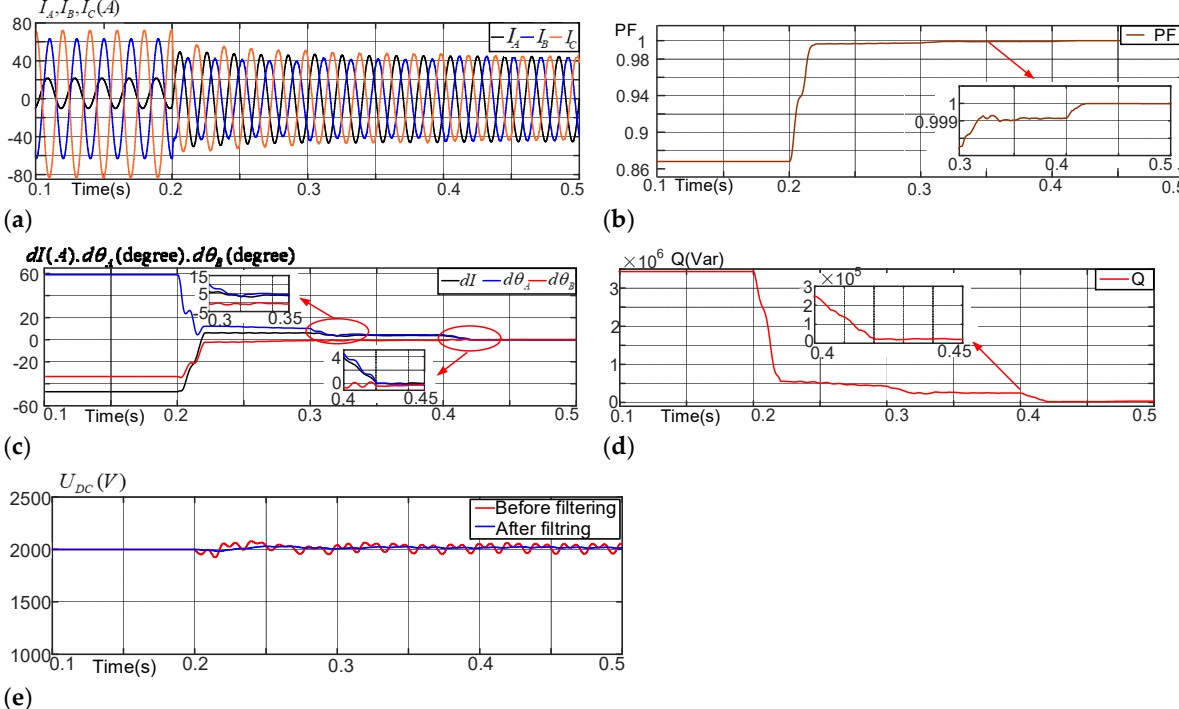

**Figure 10.** Simulation results of Case 1. (**a**) Primary side currents of V/v transformers $I_A$, $I_B$ and $I_C$; (**b**) Primary side power factor (PF) of the V/v transformer; (**c**) $dI$, $d\theta_A$ and $d\theta_B$; (**d**) Total reactive power of the power supply system $Q$; (**e**) DC capacitor voltage before and after filtering.

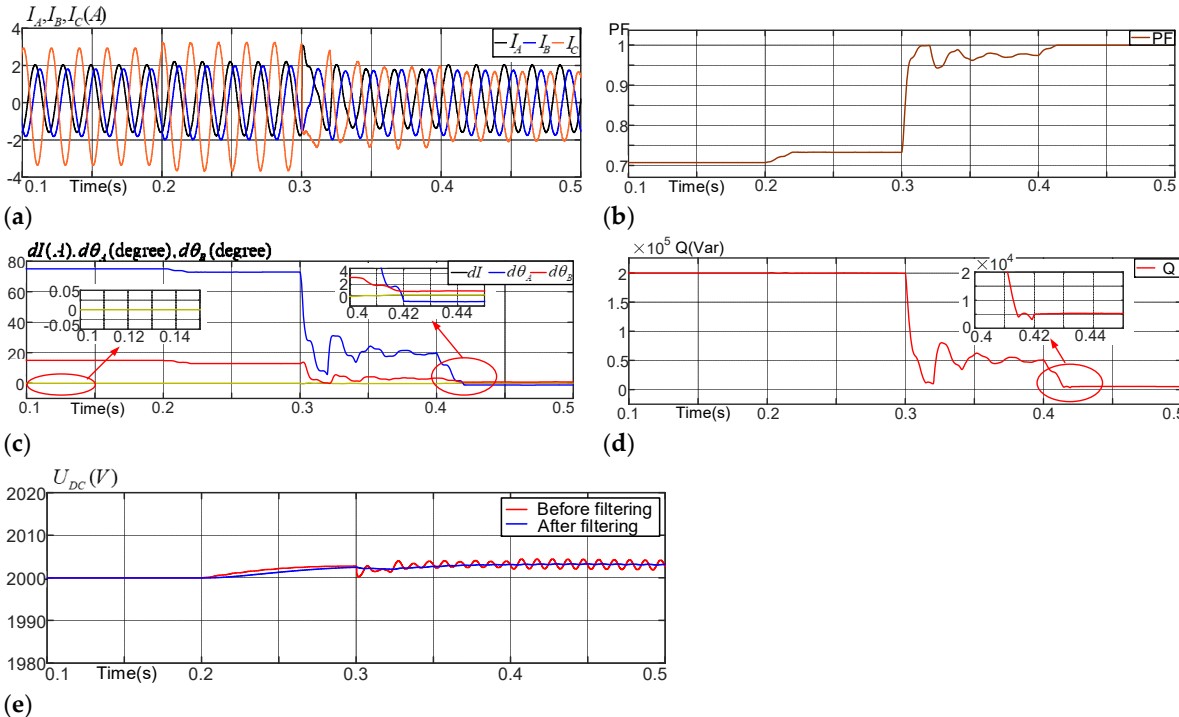

**Figure 11.** Simulation results of Case 2. (**a**) Primary side currents of V/v transformer $I_A$, $I_B$ and $I_C$; (**b**) Primary side power factor (PF) of the V/v transformer; (**c**) $dI$, $d\theta_A$ and $d\theta_B$; (**d**) Total reactive power of the power supply system $Q$; (**e**) DC capacitor voltage before and after filtering.

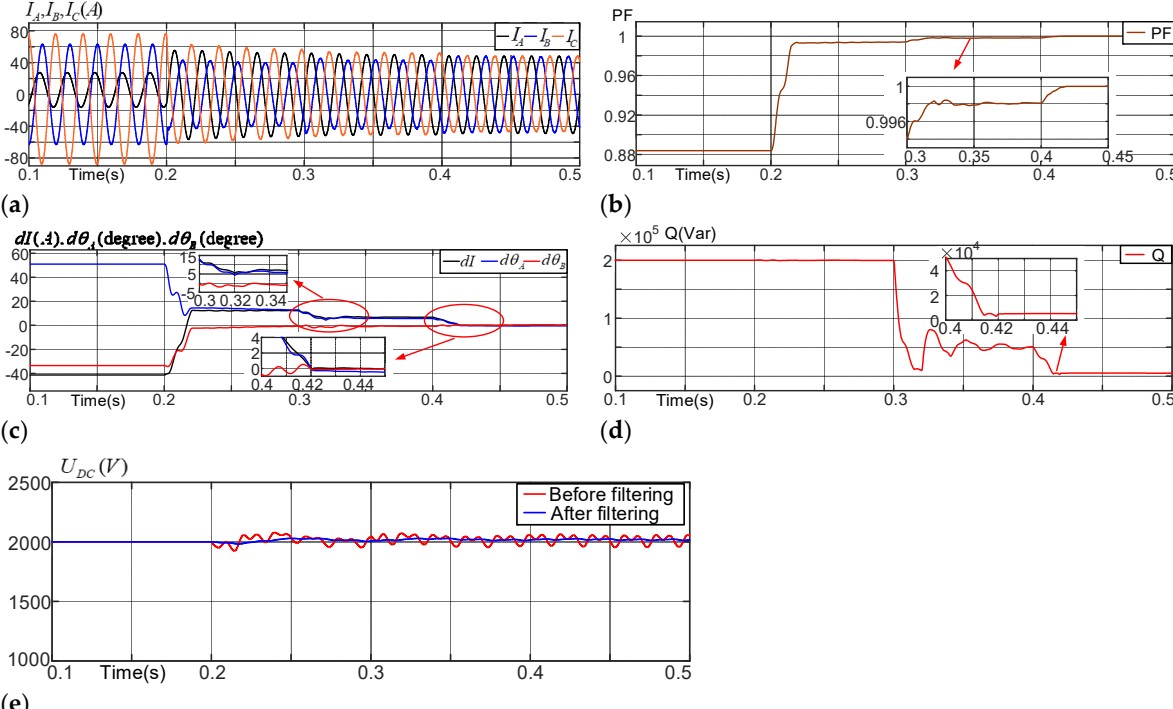

**Figure 12.** Simulation results of Case 3. (**a**) Primary side currents of V/v transformer $I_A$, $I_B$ and $I_C$; (**b**) Primary side power factor (PF) of the V/v transformer; (**c**) $dI$, $d\theta_A$ and $d\theta_B$; (**d**) Total reactive power of the power supply system $Q$; (**e**) DC capacitor voltage before and after filtering.

Additional load was added to simulate system losses in Case 3. As shown in Figure 12, the improvement effect of uncompensated RPC on power quality of power supply system was limited. Compensating RPC with a PI controller can reduce $dI$ to less than 10 A and reduce the phase differences between voltage and current by less than 10 degrees. The method proposed in this paper has a better compensation effect, which is to reduce $dI$ to less than 1 A, and the phase differences between voltage and current are also reduced to less than 1 degree.

At the same time, RPC adopts double loop control, and the DC capacitance voltage $U_{DC}$ is basically stable at 2000 V.

In order to verify the robustness of the proposed method, simulation experiments Case 4 and Case 5 were designed. Simulation parameters are shown in Table 6.

In Case 4, the PI controller, as shown in Figure 9, was used to compensate for RPC for simulation experiments under different $P_l$ conditions. Under the same conditions, the method proposed in this paper was used to compensate for RPC in Case 5.

**Table 6.** Simulation parameters of Case 4 and Case 5.

| Case | Intervals (s) | Active Power of α-Phase (kW) | Reactive Power of α-Phase (kVar) | Active Power of β-Phase (kW) | Reactive Power of β-Phase (kVar) | $P_l$ (kW) | Compensation Scheme of RPC |
|---|---|---|---|---|---|---|---|
| Case 4 | 0.1–0.2 | | | | | 100 | Without compensation |
| | 0.2–0.3 | | | | | 100 | PI controller |
| | 0.3–0.4 | 1000 | 500 | 4800 | −500 | 1000 | Without compensation |
| | 0.4–0.5 | | | | | 1000 | PI controller |
| Case 5 | 0.1–0.2 | | | | | 100 | Without compensation |
| | 0.2–0.3 | | | | | 100 | Proposed |
| | 0.3–0.4 | 1000 | 500 | 4800 | −500 | 1000 | Without compensation |
| | 0.4–0.5 | | | | | 1000 | Proposed |

The simulation results are shown in Figure 13.

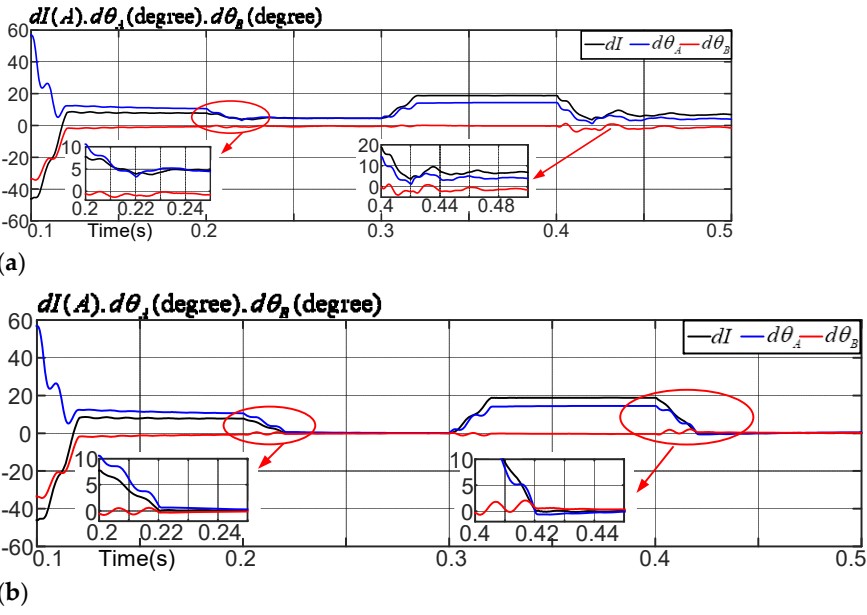

(**a**)

(**b**)

**Figure 13.** Simulation results $dI$, $d\theta_A$ and $d\theta_B$ of Case 4 and Case 5. (**a**) In Case 4; (**b**) In Case 5.

As shown in Figure 13a (0.2 s–0.3 s) and Figure 12c (0.3 s–0.4 s), when $P_l$ was small, the overshoot of $dI$, $d\theta_A$ and $d\theta_B$ was relatively small. However, when $P_l$ = 1000 kW, the overshoot of $dI$, $d\theta_A$ and $d\theta_B$ was relatively large. In Figure 13a (0.4 s–0.5 s), oscillation occurred, the adjustment time became longer and the compensation effect becomes worse. As shown in Figures 13b and 12c (0.4 s–0.5 s), the proposed method can quickly achieve the desired results and have little overshoot under different $P_l$ conditions, which indicates that the proposed method is robust.

## 5. Conclusions

This article proposes a compensation optimization scheme. First, the instantaneous reactive power theory was applied to calculate and compensate for the reactive power of the load, and then the amplitude and phase of the primary current of the V/v transformer were introduced as a reference. Secondly, using the fuzzy control to fine-tune the compensation of the system effectively reduced the compensation system error caused by system loss. The simulation results show that the compensation scheme proposed in this paper is more accurate, especially when the system has no load. In general, the method proposed in this paper can reduce 95% of the total reactive power of the system, reduce 90% of the amplitude difference of the current to less than 1 A and reduce 90% of the phase difference of the voltage and current to less than 2 degrees, which means that the voltage and current are almost completely balanced. Simulation results also show that the proposed method is robust. Because the method proposed in this paper compensates for the reactive power of the load, RPC can be applied not only to high-speed railways but also to ordinary railways with low power factors, which broadens the universality of RPC.

**Author Contributions:** Conceptualization, Y.L.; Data curation, Z.T.; Formal analysis, Y.X.; Resources, Z.T.; Validation, F.Z.; Writing: original draft, Y.L.; Writing: review & editing, J.H. All authors have read and agreed to the published version of the manuscript.

**Funding:** This research was funded by Hunan Province strategic emerging industry science and technology research and major scientific and technological achievements transformation project (No. 2018GK4016).

**Data Availability Statement:** Data is contained within the article.

**Acknowledgments:** The authors would like to acknowledge the research support from the Hunan Province strategic emerging industry science and technology research and major scientific and technological achievements transformation project (No. 2018GK4016).

**Conflicts of Interest:** The authors declare no conflict of interest.

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
