# Peer review of "Feedforward Compensation of Railway Static Power Conditioners in a V/v Traction Power Supply System"

_electronics, doi:10.3390/electronics10060656_

Round 1

Reviewer 1 Report

The work proposing a compensation optimization method, instantaneous reactive power was in theoretical and processed to calculate reactive power of the load. Amplitude and Phase current of primary side transformer made as a reference value. A Fuzzy technique is used for controlling scheme. So the methedology of this process represented here is very clear.

But I have a serious concern on result section, which is very important to validate this proposed method, this manuscript has a serious lag in simulation result section.

  1. The case 1, & Case 2 with load and Case 3 without load, this explanation is not enough to justify your proposed work.
  2. How the authors are claiming that the power factors are balanced?
  3.  Finally their is no clear information on satisfying the compensation requirements of the system.

Reviewer 2 Report

The paper deals with an important subject, however the following issues should be solved:

1) Authors are considering a known RPC power structure and compensation method. However, it has been shown elsewhere that the Hybrid Railway Power Compensator (HRPC) is more advantageous. Authors should defend why have they chosen RPC instead of HRPC.

2) There are some problems with the English language, for example:

"...In fact, the power factor of the load fluctuates, even are low in some cases."

"The RPC, consist of two converters that are connect back to back through dc capacitor with a stable dc-link voltage, is connected to the two power supply arms via..."

"Figure 3 and Figure 6 compensate most of the reactive power and active power of..."  (note:  figures do not compensate anything, instead write : Figures show block diagrams to compensate....)

Please revise all the text, checking the correctness and aproppriatness of each sentence.

3) The sentence "For convenience, the transformer connected to the Phase A and Phase B side is named the α phase power supply arm, the other side is named β phase power supply arm." seems not in agreement with  the figure, and may be is mixed. The transformer has 4 windings, Therefore, it is necessary to say the "transformer primary winding connected to the Phase A...". Besides it appears that the connection to Phase A and Phase B side do not exist and in the secondary  α phase and β phase are interchanged and mixed. Please check and clarify.

4) The RPC has  a strong drawback, relatively to HRPC, in the DC capacitor of 20000μF  at a voltage of 2000V. This is a too big capacitor bank, with low lifetime and should be avoided. Please justify why such big capacitor is needed, since there are other solutions available.

5) Most of the presented operating principle is well known (Many researchers use equation (9)....) the contribution from authors only appears in section "2.4. Problem statement", stating the issues to address. The novel contributions should be named also in the abstract and introduction.

6) "The compensation method proposed in this paper is actually a feedforward control." This approach will lead to errors when the parameters of the real system are not exactly the simulated parameters (which is always the case). Also the algorithm for current compensation is calculated in open-loop. Authors must discussed and solve these issues.

7) Authors should discuss why using a fuzzy controller instead of for example a look-up table, since the presented fuzzy rules are somewhat linear with limiting values.

8) A reference for "The defuzzification algorithm uses gravity method,..." should be provided.

9) Results shown in fig. 7 should have axis legends and units for the represented variables, and the shown waveforms on the right (b), d), f)) should also have labels.

10) Simulation results in fig.7 must be confirmed by experimental results.

11) Discussion of the results should be improved. Results obtained using the fuzzy feedforward should be compared to RPC without compensation and to RPC with the compensation but using linear controllers.

12)  Conclusions should contain the quantitative improvements the proposed compensation method brings over existing methods.

Round 2

Reviewer 1 Report

  1. Fig 2 (a) & Fig 2 (b) part of the figure was not visible in the manuscript.

The author addressed all the queries and the updated manuscript needs little modification with reference to fig. 2 (a) and fig. 2(b)

Reviewer 2 Report

Authors have improved the paper. I thank their responses. However, i do not agree with all the explanations presented:

1) Fig. 1 and the analysis is still inconsistent as the alfa voltage supplies a beta current that is divided in two alfa components, and the the beta voltage supplies an alfa current that.... Please explain or denote correctly the variables. If the notation is wrong authors are not caring enough about their paper.

2) The pdf version contains fig. 2 repeated 3 times (at least in my acrobat DC viewer), some  with cuts to the right part of the fig.. Authors must check the pdf for correctness before submitting it. If the allowed time for corrections is not enough, please ask for more days or weeks or months, but submit a proper document to a reviewer, as reviewers have also their own papers to publish...

3) There are problems with indexing as several "Error! Reference source not found.."  are effectively found in the paper. Please correct.

4) Text and figures in the pdf are located excessively to the right part of the page. Please center the text/figures in the page.

5) Experimental results should be added as simulation results are not clear, capacitors are too big and voltage ripple seems to be influencing the results.

6) Paper novelty is still in the low side. Fuzzy is a well known control method. It is said fuzzy is robust, but not proved. Try to make experiments that can prove that robustness. Try to add value to the paper.

7) Please ask for at least 3 months to have the experiments done before resubmitting the paper. The paper is not acceptable without experimental results.

Round 3

Reviewer 2 Report

The editorial board should recommend authors to present experiments and give authors time to allow the experiments

This manuscript is a resubmission of an earlier submission. The following is a list of the peer review reports and author responses from that submission.